# Multidimensional gender discrimination in workplace and depressive symptoms

**Gaeul Kim**[1], **Jinmok Kim**[1], **Su-Kyoung Lee**[2], **Juho Sim**[2], **Yangwook Kim**[3], **Byung-Yoon Yun**[4], **Jin-Ha Yoon**[3,4]*

1 Yonsei University College of Medicine, Seoul, Republic of Korea, 2 Research affairs of Yonsei University, Seoul, South Korea, 3 The Institute for Occupational Health, Yonsei University College of Medicine, Seoul, Korea, 4 Department of Preventive Medicine, Yonsei University College of Medicine, Seoul, Korea

* flyinyou@yuhs.ac

## Abstract

### Background

Discrimination is associated with depressive symptoms and other negative health effects, but little is known about the mental health risks of workplace gender discrimination. We aimed to investigate the association of workplace gender discrimination and depressive symptoms among employed women in South Korea.

### Methods

The 6th wave (2016) survey datasets of the Korean Longitudinal Survey of Women and Family (KLoWF) were analyzed for 2,339 respondents who are identified as wage workers. Depressive symptoms were evaluated by the short-form (10-item) Center for Epidemiological Studies-Depression scale. Association of workplace gender discrimination and depressive symptoms was assessed using multivariate logistic regression, adjusted for potential confounding variables including age, income satisfaction, education level, marital status, and currently diagnosed disease. We then measured the age effect using age stratification multivariate logistic regression model.

### Results

Women who experienced gender discrimination at workplace had higher odds of depressive symptoms regardless of the type of the discrimination including hiring, promotion, work assignments, paid wages, and firing. These associations were consistent in younger women below 40 years of age in regard to hiring, promotion, paid wages and firing, whereas inconsistent among older women above 40 years of age.

### Limitations

We did not investigate the effect of workplace gender discrimination on depressive symptoms in a longitudinal manner.

**Data Availability Statement:** The data underlying this study have been uploaded to figshare and are accessible using the following links: https://figshare.com/s/bab36d61326b81f7407c and https://figshare.com/s/fa94d08a85b9f9500e54.

**Funding:** This work was supported by Korea Health Industry Development Institute through "Social and Environmental Risk Research" funded by Ministry of Health & Welfare (HI19C0052). JHY had been awarded that grant. The funders had no role in study design, data collection and analysis, decision to publish, or preparation of the manuscript.

**Competing interests:** The authors declare no conflicts of interest.

## Conclusions

Workplace gender discrimination was found to be significantly associated with depressive symptoms after adjustment for socio-demographic factors. Further, women under 40 years of age were especially vulnerable to workplace gender discrimination.

## Introduction

The prevalence of depression is high throughout the world, thus posing huge economic burdens for nearly all developed and developing countries [1]. The 12-month prevalence estimate of major depressive episodes averaged 3.2% in healthy participants and 9.3% to 23.0% in participants with comorbid physical disease in the WHO World Health Survey across 60 countries[2]. Depression is associated with an increased risk of morbidity, including both cardiovascular [3] and Parkinson's disease[4]. Further, depression itself is related to diminished social functioning that results in decreased work productivity [5] and declined cognitive performance in the areas of memory, executive function, and processing speed [6].

Major depression is more likely to occur in patients with specific biological and sociopsychological risk factors. From the biological aspect, it is known that age and sex are associated with increased depression risks. For instance, the prevalence of depression is approximately two times higher in females when compared to males [7] and significantly increases with age [8]. General medical disorders, such as the neurologic [9] and metabolic [10], also increase the risk of depression. Indeed, it is more common in older adults who are living under primary care supervision and/or suffering from a wide range of medical disorders [11, 12]. Genetic factors [13], low birth weights [14], and immune related therapy involving the interferon and glucocorticoid systems [15, 16] are also associated with depression.

Regarding the sociopsychological factors, research has found that marital problems [17], low education levels, and lower incomes [18] are well-known risk factors for depression. Several studies have also reported that stressful life events increase the risk of depression [19, 20], including the loss of loved ones, sustained medical disability, and business failure. The risk of depression dramatically increases due to life events entailing long-term contextual threats. For instance, experiences of humiliation, entrapment [21], targeted rejection, and social exclusion [22] are likely to have enduring negative impacts related to depression. Although "stressfulness" can be highly subjective, perceived discrimination is also understood as a significant life stressor [23].

Discrimination is defined as being treated unfairly in any field of public life based on one's personal characteristics, such as race, gender, or religion [24]. When a group of people is stigmatized based on their characteristic, since stigma is linked to social difference, they could easily face the difficulties of discrimination. The population can be minority immigrant, lesbian, gay, bisexual, transgender, and Queer (LGBTQ), people who are overweight, people with health problems (e.g. AIDS or mental illness), or female gender. Discrimination is found in wide situations, such as social isolation of minority immigrant youth[25], experience of being bullied in adolescents with obesity[26], housing discrimination based on sexual orientation [27], or employment discrimination against people with AIDS[28].

Researchers have generally found that experiences of discrimination are harmful to health in several ways. For instance, previous studies have shown that perceived discrimination is strongly associated with poor indicators of both mental and physical health, including anxiety [29], hypertension, heightened stress responses [23], and self-reported health status [30].

Among people who had experienced discrimination due to their HIV status, internalized stigma significantly predicted cognitive-affective depression[31]. Weight-related perceived stigma as well as self-stigma is shown to be associated with psychological distress[32]. Recent studies have also shown that gender-based discrimination also produce deleterious health impacts such as cardiovascular disease [33], may increase drinking and smoking behaviors [34], and can aggravate depressive symptoms [35].

To date, most related studies have emphasized the negative health effects of discrimination based on race or ethnicity in the Western context [36–38]. However, few studies have focused on workplace gender discrimination. As such, more research is needed to determine the health effects of gender discrimination in Asian countries, especially considering that a relatively high proportion of women are unfairly treated in regard to paid wages and during the hiring process in these areas [30]. There is specifically a growing rate of depression and higher prevalence of such discrimination at workplaces throughout South Korea [39]. As such, this study investigated the association between depressive symptoms and workplace gender discrimination among women in South Korea in terms of hiring, promotion, paid wages, work assignments, training opportunities, and firing. We also examined the effects of age on workplace gender discrimination in regard to the odds of developing depression.

## Methods

### Ethics approval and consent to participate

The data lacks individual information; therefore, informed consent was not needed for the current study. The data used in this study lacks personal information. The Institutional Review Board (IRB) of the Yonsei University Health System approved the current study design (Y-2019-0176).

### Data collection and participants

In this study, we used a sample derived from the 6th wave (2016) survey datasets of the Korean Longitudinal Survey of Women and Family (KLoWF), which was conducted by the Korean Women's Development Institute (Seoul). The original KLoWF study population was randomly selected using a stratified multistage sampling design. It included a total of 9,997 adult women between 19 and 64 years of age who resided in urban and rural areas across South Korea. This study used data from the 6th wave ($N = 7,355$) because it was the most recent. The survey comprised of three major areas: family, work, and daily life. The panel questionnaire focused on sociodemographic variables, health issues, wage, and work discrimination.

Computer-assisted face-to-face interviews were also conducted. All participants provided informed consent prior to participation. Further, the KLoWF is part of a national public database that includes an identification number for each participant (available at: https://klowf.kwdi.re.kr/portal/mainPage.do). However, these identification numbers are not associated with any personal information, thus providing confidentiality. The following inclusion/exclusion criteria were implemented: (1) only wage workers ($n = 2,498$) were included from the total sample ($N = 7,355$), while (2) 159 participants were excluded due to missing values for gender discrimination at work, marital status, and/or education level. As such, data from a total of 2,339 participants were used for analysis.

### Study variables and measurements

Workplace gender discrimination was assessed according to questionnaire responses on the six following discrimination types: Hiring, promotion, paid wages, work assignments, training

opportunities, and firing. The questionnaires were as follows: (1) Hiring: If candidates have similar qualifications for appointment, men are preferred to women. (2) Promotion: Even with identical or similar careers, male workers are promoted faster than female counterparts. (3) Paid wages: Even in identical or similar positions, male workers receive higher wages and bonuses than female workers. (4) Work assignments: Duties are fixed or customarily divided between men and women. (5) Training opportunities: Even with similar duties, men have more opportunities of receiving education and training than women. (6) Firing: In case of restructuring, female workers are more likely to be forced to quit. Participants were specifically asked whether they had experienced any of these types according to a response scale for each item ranging from "never," "rarely," "almost," to "always." These responses were used to place participants into one of two categories; those who selected "never" or "rarely" were placed into the non-gender discrimination group, while those who answered "almost" or "always" were placed into the gender discrimination group.

We used the short-form (10-item) Center for Epidemiological Studies-Depression (CES-D 10) scale to assess depressive symptoms as a dependent variable. The CES-D 10 is a screening tool used to determine whether respondents experienced depressive symptoms during the week immediately prior to answering. CES-D 10 has been proven as a reliable alternative to the original CES-D 20(Kappa = 0.82, P<0.001) in classifying participants with depressive symptoms (sensitivity 91%, specificity 92%, positive predictive values 92%)[40]. Among the 10 total items, two (items 5 and 8) assess positive symptoms, while the rest focus on negative symptoms related to depression. All items are answered by selecting one of four response categories indicating the frequency of depressive moods or symptoms. A score of zero indicates that these were experienced less than once during the previous week, while a score of 1 indicates 1–2 days, 2 signifies 3–4 days, and 3 signifies more than 5 days. After reverse-scoring items 5 and 8, a total score based on all 10 items then serves as the outcome variable. Here, scores may range from 0 to 30, with higher scores indicating greater symptoms; a cut-off score of 10 is indicative of significant depressive symptoms. As such, this study used the standard cut-off score of 10 to categorize participants as having depression [40].

Age (i.e., in brackets of 19–30, 30–39, 40–49, 50–59, and 60–64), income satisfaction, education level, marital status, and health were included as covariates. Participants were asked to assess their subjective economic status, which was used to reflect their level of satisfaction with paid wages. This was answered on a 5-point scale consisting of "highly dissatisfied," "dissatisfied," "neither dissatisfied nor satisfied," "satisfied," and "highly satisfied." Answers were trichotomized for income satisfaction (i.e., "dissatisfied," "neither dissatisfied nor satisfied," and "satisfied"). Educational attainment was classified as either having completed "elementary school or less," "middle school," "high school," or "college or more." Marital status was categorized into one of four groups (i.e., "single," "never married," "married," "divorced or legally separated," or "widowed"). Currently diagnosed diseases were considered representative of individual health. Here, participants were asked whether they were currently diagnosed with heart disease, cerebrovascular disease, musculoskeletal disease, respiratory disease, gastrointestinal disease, neurologic problems, traumatic injury, or any other disease. Responses were dichotomized as "Yes" for current diagnoses (one or more diseases) and "No" for those without current diagnoses at the time of the survey.

## Statistical analyses

We first calculated the frequencies and percentages of participant characteristics and compared them to each categorized variable. We then descriptively examined the different percentage distributions of the variables of interest between the gender-discrimination and non-

gender discrimination groups. In the next step, we calculated the differences in depressive symptoms according to each variable (i.e., age, income satisfaction, education level, marital status, and currently diagnosed disease) using chi-square tests. We also checked for differences in the risk of depressive symptoms based on each type of workplace gender discrimination.

Multivariate logistic regression models revealed an odds ratio (OR) with a 95% confidence interval (CI) for depressive symptoms. We specifically employed the two following models: Model 1 (crude) and Model 2 (adjusted for age, income satisfaction, education level, marital status, and currently diagnosed disease). Further, age stratification multivariate logistic regression analyses were conducted for the below 40, 40–49, and 50 and over age groups.

## Results

Table 1 shows the characteristics of the study population according to gender-discrimination group. Mean participant age (and corresponding standard deviation/SD) was 45.0 (±11.76) years. A total of 45.7% of all participants held bachelor's degrees or higher, while 28.4% said they were satisfied with their income levels (19.04% were dissatisfied), more than half (66.65%) were married, and 13.5% had current disease diagnoses.

Nearly 30% of participants in all age groups (i.e., 33.7% in $< 30$, 29.4% in 30–39, 30.9% in 40–49, 29.6% in 50–59, and 26.1% in $\geq 60$) responded that gender discrimination existed at their workplace. Further, there were no significant differences for this issue in regard to income satisfaction (31.3% of satisfied, 30.0% of dissatisfied), education level (30.3% of $\leq$ Elementary school, 27.6% of $\geq$ College), marital status (30.0% of married, 31.5% of divorced or legally separated), or currently diagnosed diseases (26.9% of Yes, 30.6% of No).

Fig 1 shows a breakdown of participants who responded that gender discrimination existed at their workplace. Among the six types of workplace gender discrimination asked about on the survey, the most common type was related to work assignments (19.6%), followed by paid wages (16.2%), promotion (15.2%), hiring (14.7%), and training opportunities (12.18%). Finally, 14.0% responded that there was gender discrimination based on firing practices.

Table 2 shows the prevalence of depression based on demographic characteristics and the type of workplace gender discrimination. Older employees were more likely to be depressed than younger employees (9.8% vs. 27.2% for the $< 30$ vs. $\geq 60$ groups, respectively; $p < 0.001$). Participants who were less satisfied with their income levels were also more depressed ($p < 0.001$). On the other hand, the prevalence of depression was lower for the highly educated ($p < 0.001$). Marital status was also strongly associated with depressive symptoms. For instance, participants who were divorced, legally separated, or widowed more frequently expressed depressed than the single or married participants ($p < 0.001$). Finally, the prevalence of depression increased for those with current disease diagnoses ($p < 0.001$).

Except for gender discrimination related to training opportunities ($p = 0.06$), all types were associated with increased depressive symptoms ($p < 0.05$). Specifically, the prevalence rates were 23.5% vs 15.7% for hiring, 21.6% vs 16.0% for promotion, 20.6% vs. 16.1% for paid wages, 20.5% vs. 16.3 for work assignments, and 23.2% vs 15.8% for firing.

The results of the logistic regression analysis (as OR and 95% CIs) on the association between depressive symptoms and the existence of workplace gender discrimination are shown in Table 3. Crude model (a univariate logistic regression model) only considered depressive symptoms and the existence of workplace gender discrimination and was used as a baseline. For the gender-discrimination group, the ORs (95% CIs) for depressive symptoms were 1.66 (1.25–2.18) for hiring, 1.45 (1.09–1.92) for promotion, 1.35 (1.03–1.78) for paid wages, 1.34 (0.98–1.83) for training opportunities, and 1.61 (1.21–2.14) for firing. After adjusting for age, income satisfaction, education level, marital status, and current diseases, the ORs

**Table 1. Characteristics for the study population.**

| | Total | | Gender discrimination group | | Non-gender discrimination group | |
|---|---|---|---|---|---|---|
| | N | (%) | N | (%) | N | (%) |
| Total | 2339 | (100) | 705 | (30.14) | 1634 | (69.86) |
| Age (Mean = 45.0, SD = 11.76) | | | | | | |
| <30 | 306 | (13.1) | 103 | (33.7) | 203 | (66.3) |
| 30–39 | 381 | (16.3) | 112 | (29.4) | 269 | (70.6) |
| 40–49 | 849 | (36.3) | 262 | (30.9) | 587 | (69.1) |
| 50–59 | 520 | (22.2) | 154 | (29.6) | 366 | (70.4) |
| ≥60 | 283 | (12.1) | 74 | (26.1) | 209 | (57.4) |
| Income satisfaction | | | | | | |
| Satisfied | 664 | (28.4) | 208 | (31.3) | 456 | (68.7) |
| Neither satisfied nor dissatisfied | 1218 | (52.1) | 360 | (29.6) | 858 | (70.4) |
| Dissatisfied | 457 | (19.5) | 137 | (30.0) | 320 | (70.0) |
| Education level | | | | | | |
| ≤Elementary school | 178 | (7.6) | 54 | (30.3) | 124 | (69.7) |
| Middle school | 209 | (9.0) | 61 | (29.2) | 148 | (70.8) |
| High school | 882 | (37.7) | 295 | (33.4) | 587 | (66.6) |
| ≥College | 1070 | (45.7) | 295 | (27.6) | 775 | (72.4) |
| Marital status | | | | | | |
| Single, never married | 471 | (20.1) | 471 | (31.2) | 324 | (68.8) |
| Married | 1559 | (66.7) | 1559 | (30.0) | 1092 | (70.0) |
| Divorced or legally separated | 149 | (6.4) | 149 | (31.5) | 102 | (68.5) |
| Widowed | 160 | (6.8) | 160 | (27.5) | 116 | (72.5) |
| Currently diagnosed disease | | | | | | |
| Yes | 316 | (13.5) | 85 | (26.9) | 231 | (73.1) |
| No | 2023 | (86.5) | 620 | (30.6) | 1403 | (69.4) |

(95% CIs) were 1.88 (1.41–2.51), 1.68 (1.25–2.25), 1.40 (1.04–1.87), 1.39 (1.06–1.82), 1.48 (1.07–2.05), and 1.85 (1.38–2.49), respectively.

We then examined the age effects of workplace gender discrimination on the odds of developing depression (Fig 2). More than half of all workplace discrimination areas showed increased ORs (95% CIs) for depressive symptoms in the < 40 age group. Specifically, the ORs (95% CIs) for this group were 4.46 (2.58–7.68) for hiring, 3.01 (1.7–5.25) for promotion, 2.12 (1.34–3.69) for paid wages, and 3.2 (1.8–5.59) for firing. There was no statistically significant increase for the 40–49 age group. Nonetheless, the age-based effects of workplace gender discrimination and depressive symptoms increased for the ≥ 50 age group for two types of discrimination (i.e., the ORs [95% CIs] 1.63 [1.02–2.56] for hiring and 1.61 [1.01–2.58] for firing).

## Discussion

This cross-sectional study of employed women in South Korea found a statistically significant association between workplace gender discrimination and depressive symptoms. Here, the gender-discrimination group (i.e., those that answered positively for gender discrimination in their workplace) showed increased odds of depressive symptoms regardless of the type of the discrimination in five areas, including hiring, promotion, work assignments, paid wages, and firing. These associations maintained their effects even after adjusting for depression-based vulnerability controls, including age, income satisfaction, education level, marital status, and

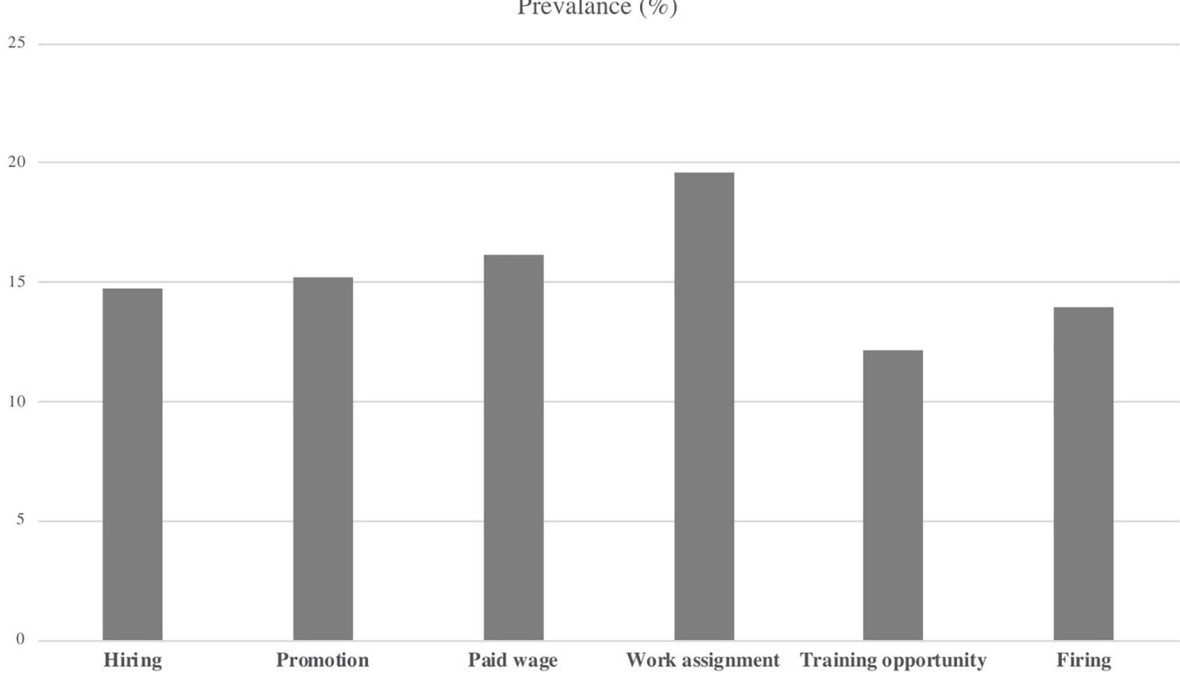

**Fig 1. Prevalence of gender discrimination at work.**

disease. Discrimination related to training opportunities was also associated with depressive symptoms after these adjustments. This study also found that younger women (those below 40 years of age) who experienced workplace gender discrimination had greater odds of developing depressive symptoms than older women (those above 40) who had experienced workplace gender discrimination. Among the younger workers, those who experienced gender discrimination in regard to hiring, promotion, paid wages, and firing had the greatest odds of developing depressive symptoms. However, these relationships were inconsistent among older workers (those above 40 years of age).

The above findings are consistent with previous studies linking experiences of discrimination to poor mental health status in a worldwide context. For instance, a cross-sectional study of 644 hospital workers found that workplace discrimination occurrences, types, and frequencies were associated with depressive symptoms [41]. Schulz et al. found a positive relationship between discrimination and changes in mental health among African-American women using longitudinal models [42], while another study on low-socioeconomic status among African-American women found that gender discrimination increased the risk for poor health and low well-being by increasing one's vulnerability to individual stressors [43].

To our knowledge, this was the first large epidemiological study to show a relationship between workplace gender discrimination and depressive symptoms among adult female workers in South Korea. These findings are especially significant because the associations persisted regardless of sociodemographic factors. Further, no other studies have stratified the association between workplace gender discrimination and depressive symptoms according to age. While some previous studies have focused on the health effects of racial discrimination in Western countries and/or general workplace discrimination, we specifically found that workplace gender discrimination was especially associated with an increased risk of depressive symptoms among younger women (those under 40 years of age).

**Table 2. Association between workplace gender discrimination and depressive symptoms.**

|  | Depressed (n = 314) | | Non-depressed(n = 1526) | | |
|---|---|---|---|---|---|
|  | n | (%) | n | (%) | p-value |
| **Age(year)** |  |  |  |  | <0.001 |
| <30 | 30 | (9.8) | 276 | (90.2) |  |
| 30–39 | 41 | (10.8) | 340 | (89.2) |  |
| 40–49 | 135 | (15.9) | 714 | (84.1) |  |
| 50–59 | 111 | (21.3) | 409 | (78.7) |  |
| 60≥ | 77 | (27.2) | 206 | (72.8) |  |
| **Income satisfaction** |  |  |  |  | <0.001 |
| Satisfied | 99 | (15.0) | 565 | (85.0) |  |
| Neither satisfied nor dissatisfied | 182 | (14.9) | 1036 | (85.1) |  |
| Dissatisfied | 113 | (24.7) | 344 | (75.3) |  |
| **Education level** |  |  |  |  | <0.001 |
| < = Elementary school | 54 | (30.3) | 124 | (69.7) |  |
| Middle school | 49 | (23.4) | 160 | (76.6) |  |
| High school | 153 | (17.3) | 729 | (82.7) |  |
| ≥University | 138 | (12.9) | 932 | (87.1) |  |
| **Marital status** |  |  |  |  | <0.001 |
| Single, never married | 51 | (10.8) | 420 | (89.2) |  |
| Married | 23.9 | (15.3) | 1320 | (84.7) |  |
| Divorced or Legally separated | 54 | (36.2) | 95 | (63.8) |  |
| Widowed | 50 | (31.2) | 110 | (68.8) |  |
| **Disease** |  |  |  |  | <0.001 |
| Yes | 108 | (34.2) | 208 | (65.8) |  |
| No | 286 | (14.1) | 1737 | (85.9) |  |
| **Discrimination at work** |  |  |  |  |  |
| **Hiring** |  |  |  |  | <0.001 |
| Yes | 81 | (23.5) | 263 | (76.5) |  |
| No | 313 | (15.7) | 1682 | (84.3) |  |
| **Promotion** |  |  |  |  | 0.0088 |
| Yes | 77 | (21.6) | 279 | (78.4) |  |
| No | 317 | (16) | 1666 | (84) |  |
| **Income** |  |  |  |  | 0.0315 |
| Yes | 78 | (20.6) | 300 | (79.4) |  |
| No | 316 | (16.1) | 1645 | (83.9) |  |
| **Work assignments** |  |  |  |  | 0.0203 |
| Yes | 94 | (20.5) | 365 | (79.5) |  |
| No | 300 | (16) | 1580 | (84) |  |
| **Training opportunity** |  |  |  |  | 0.0634 |
| Yes | 59 | (20.7) | 226 | (79.3) |  |
| No | 335 | (16.3) | 1719 | (83.7) |  |
| **Firing** |  |  |  |  | <0.001 |
| Yes | 76 | (23.2) | 251 | (76.8) |  |
| No | 318 | (15.8) | 1694 | (84.2) |  |

This study's findings are also important for their political implications. That is, it is crucial to eradicate workplace gender discrimination considering its associations with increased risks for depressive symptomatology and the fact that depression increases the risks of both physical

Table 3. Association between workplace gender discrimination and depressive symptoms.

| | Depressive symptoms | | | |
|---|---|---|---|---|
| | Crude odds ratio | (95%CI) | Adjusted odds ratio | (95%CI) |
| **Discrimination type** | | | | |
| Hiring | 1.66 | (1.25–2.18) | 1.88 | (1.41–2.51) |
| Promotion | 1.45 | (1.09–1.92) | 1.68 | (1.25–2.25) |
| Paid wage | 1.35 | (1.03–1.78) | 1.4 | (1.04–1.87) |
| Work assignments | 1.36 | (1.04–1.76) | 1.39 | (1.06–1.82) |
| Training opportunity | 1.34 | (0.98–1.83) | 1.48 | (1.07–2.05) |
| Firing | 1.61 | (1.21–2.14) | 1.85 | (1.38–2.49) |

Adjusted odds ratio: adjusted for age, income satisfaction, education level, marital status, currently diagnosed disease

and mental problems while decreasing overall work productivity. The evidence ultimately indicates the importance of ensuring workplace gender equity, especially among younger female workers.

The South Korean government established the Sex Discrimination Act in 2005. This prohibited gender discrimination in areas of education, employment, and law enforcement. However, workplace gender inequity remains a serious problem in South Korea. Such discrimination has been academically documented in a variety of forms, including hiring [44], promotions [45], paid wages [46], and expulsion (i.e., being pushed out of employment or directly fired) [47]. Further, women are more likely to experience gender discrimination than men [48]. For instance, a nationally representative study revealed that 79.3% of women reported experiencing discrimination in regard to promotional opportunities, while only 3.9% of men reported the same; further, 58.2% vs 5.2% experienced this in terms of income, while 36.9% vs 3.2% experienced it in relation to hiring, and 43.3% vs 1.1% were discriminated against in firing, respectively. The South Korean gender-based wage gap is also highest among all Organization for Economic Cooperation and Development (OECD) countries (South Korea's gender wage gap in 2016 was 36.7% compared to the OECD-35 average of 13.5%) [48]. This difference may exist due to traditional attitudes regarding fatherhood/motherhood [49] and the Confucian ideology promoting male superiority [50], which has aided the

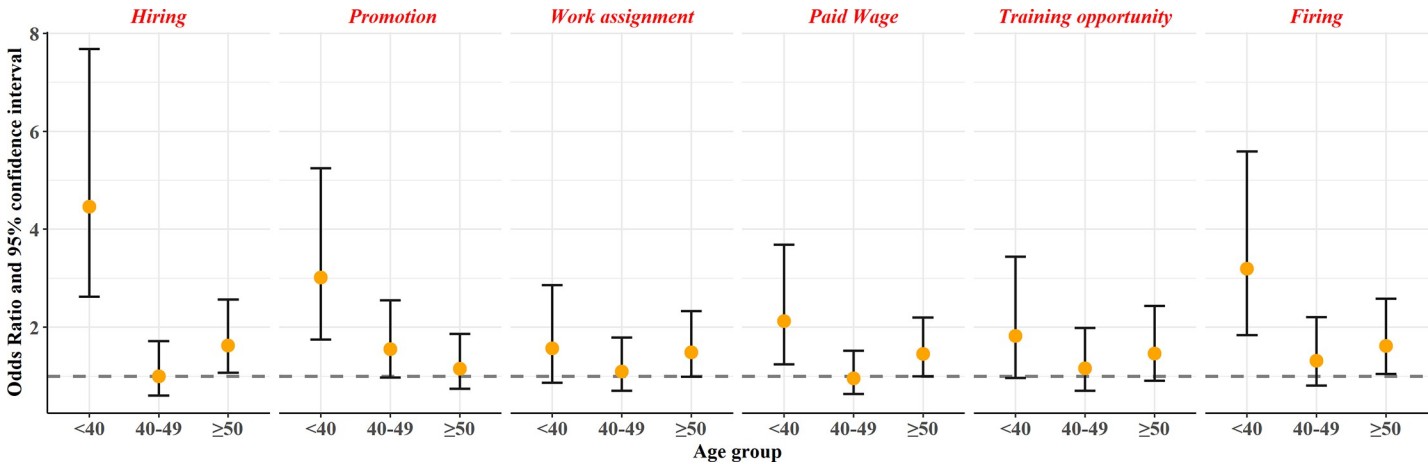

**Fig 2. Age stratified analysis of depressive symptoms and workplace discriminations.** All models were adjusted for age, income satisfaction, education level, marital status, currently diagnosed disease by multivariate logistic regression model.

establishment of a male-oriented work culture. Various policies are thus needed to completely prohibit gender discrimination while providing favorable working environments for women. Such measures may help eradicate the deep-seated workplace gender discrimination problem that currently affects the South Korean nation.

This study also had several limitations. First, it used cross-sectional association data, which cannot be used to clarify whether workplace gender discrimination precedes the depressive symptomatology or occurs as a result of certain behaviors by women who are already depressed. It is also possible that depressive symptoms may have led some individuals to report more discrimination. This makes it important to investigate all findings in a longitudinal manner. Second, there may be unrecognized confounding factors related to depressive symptoms other than workplace gender discrimination. For instance, study participants may have experienced several life events that affected their mental health status; this issue should have been considered when adjusting our data results. Lastly, we assessed workplace gender discrimination through self-reported survey data. It is thus likely that some participants were reticent to disclose experiences of gender discrimination while working at their current jobs. Further, survey responses tend to reflect personal lifelong experiences and perceptions. It should therefore be noted that one's level of perceived discrimination may differ from those of others. It is also possible that there were report and recall biases. Also, the questionnaires of current study are not validated or standardized instrument, so they did not measure objective levels of discrimination such as frequency, thereby current study results are not free from systemic error.

In conclusion, this study found that workplace gender discrimination increased the odds of depressive symptoms according to data from a nationally representative sample of employed women in South Korea. Many kinds of workplace gender discrimination were assessed, including those related to hiring, promotion, paid wages, work assignments, and firing; each of these were specifically associated with increased odds of developing depressive symptoms. Moreover, this relationship was statistically significant even after adjusting for age, income satisfaction, education level, marital status, and disease. Our findings also demonstrate that younger women (those below 40 years of age) are more vulnerable in regard to the association between workplace gender discrimination and depressive symptoms. Further study is needed to investigate the time effects of workplace gender discrimination and depressive symptoms.

## Author Contributions

**Conceptualization:** Jinmok Kim, Su-Kyoung Lee, Jin-Ha Yoon.

**Data curation:** Gaeul Kim, Jin-Ha Yoon.

**Formal analysis:** Gaeul Kim, Jin-Ha Yoon.

**Funding acquisition:** Jin-Ha Yoon.

**Investigation:** Jin-Ha Yoon.

**Methodology:** Jin-Ha Yoon.

**Writing – original draft:** Gaeul Kim, Jin-Ha Yoon.

**Writing – review & editing:** Gaeul Kim, Jinmok Kim, Su-Kyoung Lee, Juho Sim, Yangwook Kim, Byung-Yoon Yun, Jin-Ha Yoon.

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
