## [Decision Letter · Decision Letter 0]

22 Apr 2020

PONE-D-20-08368

Multidimensional Gender Discrimination in Workplace and Depressive Symptoms

PLOS ONE

Dear Dr. Yoon,

Thank you for submitting your manuscript to PLOS ONE. After careful consideration, we feel that it has merit but does not fully meet PLOS ONE’s publication criteria as it currently stands. Therefore, we invite you to submit a revised version of the manuscript that addresses the points raised during the review process.

We would appreciate receiving your revised manuscript by Jun 06 2020 11:59PM. To enhance the reproducibility of your results, we recommend that if applicable you deposit your laboratory protocols in protocols.io, where a protocol can be assigned its own identifier (DOI) such that it can be cited independently in the future. For instructions see: http://journals.plos.org/plosone/s/submission-guidelines#loc-laboratory-protocols

We look forward to receiving your revised manuscript.

Kind regards,

Amir H. Pakpour, Ph.D.

Academic Editor

PLOS ONE

2. Your ethics statement must appear in the Methods section of your manuscript. If your ethics statement is written in any section besides the Methods, please move it to the Methods section and delete it from any other section. Please also ensure that your ethics statement is included in your manuscript, as the ethics section of your online submission will not be published alongside your manuscript.

3. Please remove your figures from within your manuscript file, leaving only the individual TIFF/EPS image files, uploaded separately.  These will be automatically included in the reviewers’ PDF.

Reviewers' comments:

Reviewer's Responses to Questions

**Comments to the Author**

1. Is the manuscript technically sound, and do the data support the conclusions?

Reviewer #1: Partly

2. Has the statistical analysis been performed appropriately and rigorously? 

Reviewer #1: Yes

3. Have the authors made all data underlying the findings in their manuscript fully available?

Reviewer #1: Yes

4. Is the manuscript presented in an intelligible fashion and written in standard English?

Reviewer #1: Yes

5. Review Comments to the Author

Reviewer #1: The authors investigate an important topic on gender discrimination. Additionally, the large sample size with a good sampling method increase the external validity of the study findings. However, some revisions are needed before I recommend publication. Please see my comments below.

1. For the Introduction, please describe more thoroughly in the concept of discrimination and stigma. Specifically, I would like to see the authors mention that different populations (e.g., people with mental illness, people with overweight, people who are transgender, people who have same-sex marriage) all face the difficulties of discrimination. Then, the authors can narrow down the discrimination problem from different populations to their study topic; that is, female.

There is an ongoing special issue in the IJERPH, which collect essential information for the authors to consider and cite: https://www.mdpi.com/journal/ijerph/special_issues/stigma

The authors may also consider the following reference if relevant:

Ahorsu, D. K., Lin, C.-Y., Imani, V., Griffiths, M. D., Su, J.-A., Latner, J. D., Marshall, R. D., Pakpour, A. H. (2020). A prospective study on the link between weight-related self-stigma and binge eating: Role of food addiction and psychological distress. International Journal of Eating Disorders, 53(3), 442-450.

Alimoradi, Z., Golboni, F., Griffiths, M. D., Broström, A., Lin, C.-Y., & Pakpour, A. H. (in press). Weight-related stigma and psychological distress: A systematic review and meta-analysis. Clinical Nutrition. https://doi.org/10.1016/j.clnu.2019.10.016

Lin, C.-Y., Imani, V., Broström, A., Huus, K., Björk, M., Hodges, E. A., Pakpour, A. H. (in press). Psychological distress and quality of life in Iranian adolescents with overweight/obesity: Mediating role of weight bias internalization and insomnia. Eating and Weight Disorders. doi: 10.1007/s40519-019-00795-5

2. For the first sentence in the Introduction, the authors mentioned the high prevalence of depression worldwide. Please directly provide the prevalence for readers to have a better idea here.

3. A major problem in the present study is that it is unclear how the gender discrimination was assessed. The authors mentioned "Participants were specifically asked whether they had experienced any of these types [hiring, promotion, paid wages, work assignments, training opportunities, and firing] according to a response scale for each item ranging from “never,” “rarely,” “almost,” to “always.”" The problem here is that only reading the descriptions, I cannot observe any "discrimination" concept here. For example, if the question is "Do you have any problems in hiring?" Then, this example question is not at all "discrimination". Therefore, the authors should give detailed information on the discrimination measure, especially this is their study's core concept.

4. Please provide psychometric properties of the short version CES-D, especially how the sensitivity and specificity of the cutoff 10 in the CES-D.

5. It is unclear whether all the discrimination items (hiring, promotion, paid wages, work assignments, training opportunities, and firing) were included in the same Model 2 of logistic regression model.

6. Following my comment #5, please use AOR instead of OR for Model 2; and COR instead of OR for the Crude Model. Also, add the footnotes to explain AOR (adjusted odds ratio) and COR (crude odds ratio).

7. Page 14, "The government established the Sex Discrimination Act in 2005....." should be explicitly indicate that it is South Korean government.

8. Apart from the limitation of self-report, I think that the questionnaire on gender discrimination is not a standardized instrument. Therefore, this should also be acknowledged as a limitation.

6. PLOS authors have the option to publish the peer review history of their article (what does this mean?). If published, this will include your full peer review and any attached files.

Reviewer #1: No

---

## [Author Response · Author response to Decision Letter 0]

23 May 2020

Comment 1). For the Introduction, please describe more thoroughly in the concept of discrimination and stigma. Specifically, I would like to see the authors mention that different populations (e.g., people with mental illness, people with overweight, people who are transgender, people who have same-sex marriage) all face the difficulties of discrimination. Then, the authors can narrow down the discrimination problem from different populations to their study topic; that is, female.

There is an ongoing special issue in the IJERPH, which collect essential information for the authors to consider and cite: https://www.mdpi.com/journal/ijerph/special_issues/stigma

The authors may also consider the following reference if relevant:

Ahorsu, D. K., Lin, C.-Y., Imani, V., Griffiths, M. D., Su, J.-A., Latner, J. D., Marshall, R. D., Pakpour, A. H. (2020). A prospective study on the link between weight-related self-stigma and binge eating: Role of food addiction and psychological distress. International Journal of Eating Disorders, 53(3), 442-450.

Alimoradi, Z., Golboni, F., Griffiths, M. D., Broström, A., Lin, C.-Y., & Pakpour, A. H. (in press). Weight-related stigma and psychological distress: A systematic review and meta-analysis. Clinical Nutrition. https://doi.org/10.1016/j.clnu.2019.10.016

Lin, C.-Y., Imani, V., Broström, A., Huus, K., Björk, M., Hodges, E. A., Pakpour, A. H. (in press). Psychological distress and quality of life in Iranian adolescents with overweight/obesity: Mediating role of weight bias internalization and insomnia. Eating and Weight Disorders. doi: 10.1007/s40519-019-00795-5

(Response) Thank you for your constructive comments. We have added the concept of stigma and examples of discrimination in different populations. 

In introduction section, 

“When a group of people is stigmatized based on their characteristic, since stigma is linked to social difference, they could easily face the difficulties of discrimination. The population can be minority immigrant, lesbian, gay, bisexual, transgender, and Queer (LGBTQ), people who are overweight, people with health problems (e.g. AIDS or mental illness), or female gender. Discrimination is found in wide situations, such as social isolation of minority immigrant youth, experience of being bullied in adolescents with obesity, housing discrimination based on sexual orientation, or employment discrimination against people with AIDS.”, and

“. Among people who had experienced discrimination due to their HIV status, internalized stigma significantly predicted cognitive-affective depression. Weight-related perceived stigma as well as self-stigma is shown to be associated with psychological distress. Recent studies have also shown that gender-based discrimination also produce deleterious health impacts such as cardiovascular disease , may increase drinking and smoking behaviors , and can aggravate depressive symptoms .”

Please read our revised manuscript, too.

Comments 2). For the first sentence in the Introduction, the authors mentioned the high prevalence of depression worldwide. Please directly provide the prevalence for readers to have a better idea here.

(Response) We added sentence regarding prevalence of depression, as your comment. 

“The 12-month prevalence estimate of major depressive episodes averaged 3.2% in healthy participants and 9.3% to 23.0% in participants with comorbid physical disease in the WHO World Health Survey across 60 countries[11].”

Comment 3). A major problem in the present study is that it is unclear how the gender discrimination was assessed. The authors mentioned "Participants were specifically asked whether they had experienced any of these types [hiring, promotion, paid wages, work assignments, training opportunities, and firing] according to a response scale for each item ranging from “never,” “rarely,” “almost,” to “always.”" The problem here is that only reading the descriptions, I cannot observe any "discrimination" concept here. For example, if the question is "Do you have any problems in hiring?" Then, this example question is not at all "discrimination". Therefore, the authors should give detailed information on the discrimination measure, especially this is their study's core concept.

(Response) The exact questionnaires were as follows. And we added the sentence below in methods section. Please read our revised manuscript, too. 

In method section, 

(1) Hiring: If candidates have similar qualifications for appointment, men are preferred to women. 

(2) Promotion: Even with identical or similar careers, male workers are promoted faster than female counterparts. 

(3) Paid wages: Even in identical or similar positions, male workers receive higher wages and bonuses than female workers. 

(4) Work assignments: Duties are fixed or customarily divided between men and women. 

(5) Training opportunities: Even with similar duties, men have more opportunities of receiving education and training than women. 

(6) Firing: In case of restructuring, female workers are more likely to be forced to quit.

4. Please provide psychometric properties of the short version CES-D, especially how the sensitivity and specificity of the cutoff 10 in the CES-D.

(Response) Thank you for your comments. The CES-D is wide used psychological questionnaires. And we reviewed the article about psychometric properties of CESD, and summarized it into method section, as below. Furthermore, we added validation (such as Kappa value, sensitivity, specificity) in method section.

“CES-D 10 has been proven as a reliable alternative to the original CES-D 20(Kappa=0.82, P<0.001) in classifying participants with depressive symptoms (sensitivity 91%, specificity 92%, positive predictive values 92%)”

5. It is unclear whether all the discrimination items (hiring, promotion, paid wages, work assignments, training opportunities, and firing) were included in the same Model 2 of logistic regression model.

(Response) One of discrimination item was included in each logistic regression model (Crude). So we repeated 6 times to make Crude OR column of Table 3. Then, we adjusted variables for each logistic regression model.

6. Following my comment #5, please use AOR instead of OR for Model 2; and COR instead of OR for the Crude Model. Also, add the footnotes to explain AOR (adjusted odds ratio) and COR (crude odds ratio).

(Response 6) Thank you for your detailed comment. we used full name of each OR as “crude odds ratio” and “adjusted odds ratio” in row of Table as your comment. 

7. Page 14, "The government established the Sex Discrimination Act in 2005....." should be explicitly indicate that it is South Korean government.

(Response 7) We added the ‘South Korean government’, in that sentence. 

8. Apart from the limitation of self-report, I think that the questionnaire on gender discrimination is not a standardized instrument. Therefore, this should also be acknowledged as a limitation.

(Response 8) We agree that the questionnaire was not validate or standardized for study. So, we added sentences in limitation section as below. 

“Also, the questionnaires of current study are not validated or standardized instrument, so they did not measure objective levels of discrimination such as frequency, thereby current study results are not free from systemic error.”

---

## [Editor Report · Decision Letter 1]

27 May 2020

Multidimensional Gender Discrimination in Workplace and Depressive Symptoms

PONE-D-20-08368R1

Dear Dr. Yoon,

We are pleased to inform you that your manuscript has been judged scientifically suitable for publication and will be formally accepted for publication once it complies with all outstanding technical requirements.

With kind regards,

Amir H. Pakpour, Ph.D.

Academic Editor

PLOS ONE
---

## [Editor Report · Acceptance letter]

8 Jun 2020

PONE-D-20-08368R1 

Multidimensional Gender Discrimination in Workplace and Depressive Symptoms 

Dear Dr. Yoon:

I'm pleased to inform you that your manuscript has been deemed suitable for publication in PLOS ONE. Congratulations! Your manuscript is now with our production department. 

Kind regards, 

on behalf of

Dr. Amir H. Pakpour 

Academic Editor

PLOS ONE